# Influence of Different Pretreatments on the Structure and Hydrolysis Behavior of Bamboo: A Comparative Study

**DOI:** 10.3390/ma12162570

**Published:** 2019-08-12

**Authors:** Xuemin Qi, Jie Chu, Liangliang Jia, Anuj Kumar

**Affiliations:** 1College of Forestry, Northwest A&F University, 3 Taicheng Road, Yangling 712100, China; 2Natural Resources Institute Finland (Luke), Production Systems, Tietotie 2, FI-02150 Espoo, Finland

**Keywords:** structural properties, bamboo, hydrolysis abilities, pretreatment

## Abstract

In the present study, three pretreatments of sodium hydroxide (NaOH), sulfuric acid (H_2_SO_4_), and glycerin were employed with bamboo fibers at two different temperatures of 117 °C and 135 °C, respectively. The chemical composition and structural characterization of the pretreated bamboo fibers were comparatively studied using spectroscopic and wet chemistry methods. Furthermore, the comparative hydrolysis behaviors of pretreated bamboo were studied due to the synergistic interaction between cellulases and xylanase. The NaOH treatment increased the holocellulose contents to 87.4%, and the mean diameter of the cellulose fibers decreased from 50 ± 5 µm (raw fiber bundles) to 5 ± 2 µm. The lignin content and the degree of cellulose polymerization both decreased, while the crystallinity index of cellulose and thermostability increased. The hydrolysis yields of NaOH pretreated bamboo at 135 °C increased from 84.2% to 98.1% after a supplement of 0.5 cellulose to 1 mg protein/g dry xylan. The NaOH pretreatment achieved optimal enzymatic digestibility, particularly at higher temperatures as indicated by the results.

## 1. Introduction

Bamboo is a perennial herb with rapid growth, short retreading, and easy reproduction. It has been reported that about 30–35 million tons of bamboo are produced per year in Asia. Annual production of bamboo in China reached 1356 billion culms [1], which was mostly used in the construction, pulp and paper, food, and furniture sectors [2]. In recent times, the bioconversion of bamboo to produce bioethanol became an important research focus [3,4,5]. Pretreatment plays a key role in disrupting the recalcitrance of lignocellulose to achieve efficient conversion into bioethanol. Effectively removing lignin by the pretreatment process improves subsequent enzymatic digestibility of the bamboo [6,7,8]. Previous studies have shown that bamboo is difficult to pretreat compared with other agricultural materials due to the high degree of lignification and complex cell-wall polysaccharide fiber constituents [9,10]. More importantly, the effects of different pretreatments on the bamboo constituents and structural properties have not been elucidated. Li [1,2] reported that an acid pretreatment was more effective for bamboo than an alkaline pretreatment, as more hemicelluloses are degraded by acids. However, Xin [11] reported that alkali effectively removes lignin by cracking the ester bonds between *p*-coumaric/ferulic acid and hemicelluloses. 

The enzymatic hydrolysis of lignocelluloses is improved by using an organic solvent, such as glycerin, as a pretreatment and solvent [12]. Romaní [13] described that a glycerol-organosolv pretreatment improves enzymatic saccharification of cellulose in eucalyptus wood. Abdulkhani [14] reported that using atmospheric glycerol auto-catalyticorganosolv to pretreated wood is an effective delignification method. 

Reducing the high input of chemicals and/or energy was essential to develop the energy-saving equipment for the pretreatment and bioconversion of biomass [15]. A sand bath can be used as a effective heating medium to control temperature, resulting in a smooth heat condition. This would avoid the degradation of carbohydrates, thereby leading to a higher yield of sugars. Furthermore, use of a sand bath has other advantages, such as faster heating and greater energy efficiency and security, than those of water and oil baths [16]. 

The main aim of this study was to study the influence of acidic (H_2_SO_4_), alkaline (NaOH), and glycerol pretreatments on the structural and hydrolysis behavior of a bamboo biomass at two different temperatures in a sand bath heating medium. The structural analysis of pretreated bamboo was by X-ray diffraction (XRD), scanning electron microscopy (SEM), Fourier transform-infrared spectroscopy (FT-IR), thermogravimetric analysis (TGA), and solid-state cross polarization/magic angle spinning (CP/MAS ^13^C NMR). The effectiveness of the pretreatment was further verified by the series of structural analyses described above. The synergistic interaction between cellulases and xylanase (XYL) during hydrolysis of the pretreated bamboo samples was also investigated. The enzymatic digestibility of the bamboo after the NaOH pretreatment was explored. The desired enzymatic hydrolysis was achieved in an acetic acid and alkaline pretreatment system with a sand bath device. The results will provide a basis for further improvements in utilizing wood resources and realize the development and utilization of biomass energy by hydrolysis saccharification.

## 2. Materials and Methods

### 2.1. Materials

Bamboo powder samples were obtained during the summer of 2018 from the United States Department of Agriculture Forestry Bureau at the Southern Pines Research Station. Prior to chemical pretreatment, the air dried bamboo was ground using a 1.0 mm hammer mill. The average moisture content of the ground dried bamboo was 6.18% (wt%). The bamboo was sealed in a zip pocket and maintained at 4 °C until use.

The chemical reagents, such as H_2_SO_4_, NaOH, NaClO_2_, toluene, methanol, ethanol, and glacial acetic acid were purchased from Fisher Scientific (Pittsburgh, PA, USA) and used as received. The pretreatment equipment included an electronic sand bath pot and a thermometer. A bamboo total sample of 100 g oven-dry basis was used for the pretreatment experiment. The commercial enzyme preparations Celluclast 1.5 L, Novozyme 188 (Novozymes A/S, Bagsværd, Denmark), xylanase preparation (CFAD-X2753-50G, Sigma, St. Louis, MO, USA), and other chemicals used in this study were purchased from Sigma.

### 2.2. Pretreatment

As shown in Figure 1, about 10 g of bamboo powder (dry weight) was pretreated with reagent (1.5% H_2_SO_4_, 10% NaOH, or 10% glycerol) and placed in a bath at 117 °C and 135 °C for 1 h [17]. The solid-liquid ratio was 1:10. After the pretreatment, the solid was washed and repeatedly centrifuged (5000 rpm, 10 min) with distilled water until the pH of the supernatant was neutral. The solids were air dried and stored for later structural and enzymatic hydrolysis analyses.

### 2.3. Components Analysis

Holocellulose, cellulose, lignin, extractive, and ash contents were determined according to the standard protocol of the Renewable Energy Laboratory [18]. A 10 g sample was extracted for 8 h in a flask reactor with a reflux condenser using 100 mL of an ethanol/benzene solvent (1:2, v/v). About 2 g of extractive free sample was taken to a 250 mL round-bottom flask to which 150 mL water was added, along with 0.2 mg glacial acetic acid, and 1 g NaClO_2_ to determine the holocellulose content. The mixture was heated slowly for 5 h and a 0.2 mg aliquot of glacial acetic acid and 1 g NaClO_2_were added every hour and cooled to room temperature. Holocellulose was obtained by filtration and drying. A 1 g holocellulose sample was placed in a 250 mL beaker to isolate cellulose and 50 mL of 17.5% NaOH was added and stirred for 30 min. Then, 50 mL of water was added, reacted for 5 min, and the residue was obtained by filtration. Furthermore, 50 mL of 8.3% NaOH and 40 mL of 10% (wt) glacial acetic were added to the residue, and the reaction was held for 3 min. Finally, cellulose was obtained by filtration, washing with water, and drying.

The ash test was carried out by weighing the biomass in a crucible at 575 ± 25 °C during the gravimetric analysis for ash content (referring to LAP #42622).

Lignin content was classified into acid-insoluble and acid-soluble lignin. The bamboo sample was pretreated with 72% sulfuric acid for 1 h, and the polymerized carbohydrate was hydrolyzed to a monomer form, which was dissolved in the hydrolyzate. The sample was separated by filtration into a filtrate and a solid fraction. The acid-insoluble residue was determined by measuring the weight of the solid portion after drying at 105 °C. The amount of acid-insoluble lignin was obtained by correcting the acid-insoluble ash content (minus the weight of the acidic end-substrate from the ash weight).

### 2.4. Average Degree Of Polymerization (DP)

The average DP of cellulose was determined in accordance with British Standards restrictions. Viscosity was determined according to Part I. The cuprite hylene diamine method (BS 6306: Part 1: 1982). Sample viscosity was determined based on cuprite hylene diamine viscosity [η], which was obtained using the following equation: (1)$${\mathrm{DP}}^{0.905}=1.65\;\left[\eta \right]$$
where DP is the average degree of polymerization, and η (mL/g) is intrinsic viscosity. The viscosity measurement instrument was a capillary viscometer (Shimadzu, Tokyo, Japan), DP = (0.80 ± 0.05) mm.

### 2.5. FT-IR Spectroscopy

A NicoletiS10 Fourier transform infrared spectrometer sold by Thermo Fisher (Waltham, MA, USA) was used under the following test conditions. The spectral range was 4000–400 cm^−1^, spectral resolution was 4 cm^−1^, and the scans were performed 64 times.

### 2.6. Analysis of Crystallinity by XRD

The crystallinity of the bamboo samples was determined by XRD using the Rigaku Ultima3 x (Louisiana State University, Baton Rouge, LA, USA) with scanning parameters of Ni-filtered Cu Kα radiation (λ= 1.54060).

This study used the Scherer formula (Segal, 1959) to calculate d and CrI, as follows:(2)$$d=\frac{\lambda }{2sin\theta }$$
(3)CrI=d×100
where d is the crystal layer spacing (nm); λ is the incident wavelength (0.154 nm); and θ is the diffraction angle.

### 2.7. Analysis of the Chemical Structure by Solid-State CP/MAS ^13^C NMR

Solid-state CP/MAS ^13^C NMR measurements were recorded at 100MHz on a Bruker AV-400 spectrometer (Billerica, MA, USA) with a wide bore magnet and a 4 mm CP/MAS probe. ^13^C CP/MAS data were collected at 20 °C with 50 kHz spectra width, 2-ms cross-polarization contact time, and 0.035-s acquisition time. The number of scans was 8196, and rotational velocity of the sample was 12 Hz. The data were processed using Bruker Top Spin Software.

### 2.8. Scanning Electron Microscopy

The surface morphology of the bamboo samples was observed using SEM (JSM-6610; Jeol, Tokyo, Japan). The test sample was coated with gold using a vacuum sputter coater prior to the SEM analysis.

### 2.9. TGA

The TG and differential thermogravimetric (DTG) curves of the raw and pretreated bamboo samples were characterized through TG analysis with a TA Q50 Analyzer (TA Instruments, New Castle, DE, USA). Approximately 2 mg of sample was used for the TG/DTG analysis. The samples were heated from room temperature to 600 °C under a flow of 60 mL/min of nitrogen gas at a heating rate of 20 °C/min.

### 2.10. Enzymatic Hydrolysis 

The raw and NaOH pretreated bamboo were hydrolyzed by cellulase in tubes with 3 mL of 50 mM sodium citrate buffer (pH 5.0) at 50 °C in a shaking incubator at 200 rpm. The dry matter (DM) content of the substrate was 2%. A 0.02% NaN3 was added to the hydrolysis broth to prevent bacterial infection. The cellulase preparation (CEL) contained both the Celluclast 1.5 L and Novozyme 188 preparations; the CEL was dosed at 5–40 FPU/g DM Celluclast 1.5 Land 500 nkat/g DM Novozyme 188. The samples were hydrolyzed, and then boiled for 10 min to stop enzymatic hydrolysis. After cooling, the samples were centrifuged at 10,000 rpm for 10 min, and the reducing sugars in the supernatants were analyzed. To investigate the synergistic interaction between CEL and XYL, hydrolysis of the NaOH pretreated bamboo by CEL and/or XYL was investigated as described above. CEL was dosed at 5 and 10 FPU Celluclast 1.5 L/g DM, and 500 nkat Novozyme 188/g DM. The XYL dosage was 2 mg protein/g DM. The samples were withdrawn after 48 h and boiled for 10 min to stop enzymatic hydrolysis. 

The concentration of glucose and xylose in the supernatants was determined using the Agilent 1260 infinity high performance liquid chromatography system (Agilent Technologies, Palo Alto, CA, USA). This system was equipped with a refractive index detector and an autosampler. An ion-moderated partition chromatography column (Aminex column HPX-87H) with a Cation H micro-guard cartridge was used. The column was maintained at 45 °C with 5-mM H_2_SO_4_ as the eluent at a flow rate of 0.5 mL/min. Before injection, the samples were filtered through 0.22-µm Micro-PES filters, and a volume of 20 µL was injected. Peaks were detected by refractive index and were identified and quantified by comparison to the retention times of authenticated standards (D-glucose and D-xylose).

The glucose and xylose yields from the hydrolysis of samples were calculated using the following equations:
(4)$$\mathrm{Cellulose}\;\mathrm{to}\;\mathrm{glucose}\;\mathrm{conversion}\;(\%)\;=\frac{\mathrm{Glucose}\;\mathrm{released}*0:9}{\mathrm{Theoretical}\;\mathrm{amount}\;\mathrm{of}\;\mathrm{cellulose}\;\mathrm{in}\;\mathrm{substrates}}\;\times \;100$$
(5)$$\mathrm{Xylan}\;\mathrm{to}\;\mathrm{xylose}\;\mathrm{conversion}\;(\%)\;=\frac{\mathrm{Xylose}\;\mathrm{released}*0.88}{\mathrm{Theoretical}\;\mathrm{amount}\;\mathrm{of}\;\mathrm{Xylanin}\;\mathrm{substrates}}\;\times \;100$$

## 3. Results and Discussion 

### 3.1. Chemical Composition of Untreated and Pretreated Bamboo

The chemical composition of the raw material and pretreated bamboo samples is summarized in Table 1. Untreated bamboo contained 60.45% holocellulose (43.76% cellulose and 17.69% hemicellulose) 25.96% lignin, and 11.54% extractives; these results agreed with previous studies [19]. Holocellulose and cellulose contents increased significantly from 60.45% and 43.76% to 87.38% and 73.94%, respectively after the 135 °C NaOH pretreatment. The NaOH pretreatment resulted in higher holocellulose and cellulose contents compared with the H_2_SO_4_ and glycerol pretreatments. In addition, the higher pretreatment temperature resulted in higher holocellulose and cellulose and lower lignin content. These results are in good agreement with previous studies reporting that an alkaline (NaOH, 0.125 M) pretreatment is effective for delignification of pinecones and 72% holocellulose was generated. The quantity of lignin decreased after pretreatment, and the order of lignin content was glycerol > H_2_SO_4_ > NaOH pretreatment. Relatively low lignin content may have occurred after the NaOH pretreatment because the effective lignin glass transition efficiently solubilized lignin, which was in accordance with a previous result that total lignin content decreases to 22.4%, with 13.7% acid soluble lignin and 8.7% Klason lignin after a dilute acid pretreatment. The acid and alkali pretreatments decreased the ash content but not the glycerol pretreatment. The large amount of ash after glycerol pretreatment may represent some ash and non-structural carbohydrates, starch, wax, including fat in glycerin, tannins, and pigments [20].

### 3.2. Degree of Polymerization

The viscosities of the samples were determined and the related parameters were calculated to investigate the cellulose depolymerization behavior (Table 2). The η and DP values were calculated in the ranges of 103.88 to 132.62 and 336 to 396, respectively for the different pretreatments. The high DP value of the native sample was mainly due to the low deformation value of the cellulose crystal lattice. Relatively low DP values were obtained after the different pretreatments, probably because the cellulose molecules broke and became shorter, resulting in a lower DP [21].The DP value after pretreatment decreased slightly compared with that of native bamboo from 396 to 329 [22]. The H_2_SO_4_ pretreatment resulted in the highest DP value (372), followed by glycerol (362) and the NaOH pretreatment (329). The lowest DP value after the NaOH pretreatment was mainly due to deformation of the cellulose lattice during alkaline processing. The higher pretreatment temperature caused lower DP values for both the H_2_SO_4_ and NaOH pretreatments.

### 3.3. Analysis of the Bamboo Samples by FT-IR Spectroscopy

The FT-IR spectra in the 4000–400 cm^−1^ range and the changes in the adsorption peaks of the functional groups are shown in Figure 2. The peak at about 896 cm^−1^ (characteristic peak of *β*-D glycosidase) in the pretreated bamboo samples was sharper than that in the untreated bamboo, indicating increased polysaccharide content after pretreatment [23]. The band at 896 cm^−1^ was more obvious after the NaOH pretreatment than the bands after the H_2_SO_4_ and glycerol pretreatments, indicating higher polysaccharide contents in the NaOH pretreated bamboo. These results were in accordance with previous data (Table 1). In addition, the band shift from 1480 cm^−1^ to 1482 cm^−1^ after the NaOH pretreatment revealed changes in the intermolecular hydrogen bonding and conversion of the cellulose crystal structure [7]. The characteristic bands of lignin were detected at 1230 cm^−1^ and were assigned to CH_3_, COOH, and C–O stretching (Figure 2). The intensity of this peak in native bamboo indicates that native bamboo contains a high amount of lignin. After pretreatment, the absorbance peaks at 1730 cm^−1^ and 1230 cm^−1^ disappeared because of the removal of esters from the carboxylic groups [7]. These results confirm the lower lignin content after the H_2_SO_4_ and NaOH pretreatments (Table 1).

### 3.4. Analysis of the Bamboo Samples by XRD

The crystallinities of the native and treated bamboo samples were determined by XRD (Figure 3). Table 3 shows the crystallinity index (CrI) values of the untreated and pretreated bamboo samples. The CrI of the untreated bamboo was about 36.91% due to the presence of large amounts of amorphous substances, including lignin, hemicelluloses, and extractives. The treatments increased CrI from 36.91 in the raw material to 46.50. More crystalline structures were formed and the NaOH pretreatment at the higher temperature showed a higher CrI, followed by the H_2_SO_4_ and then the glycerol pretreatment. Pretreatment caused an obvious increase in the CrI, possibly because of the removal of amorphous components. The NaOH pretreatment increased the CrI of bamboo the most. Delignification resulted in an increase of CrI to different degrees compared to the corresponding native. In addition, an increase in CrI of bamboo was observed after the NaOH pretreatment, which could have resulted from preferential hydrolysis and removal of amorphous cellulose [24].

This study used the FT-IR spectroscopy results for the ratio of the absorption bands at 1428 cm^−1^ and 896 cm^−1^ and 3338 cm^−1^ and 1336 cm^−1^ to calculate the lateral order index (LOI) and hydrogen bond intensity (HBI), respectively. In general, the LOI is an empirical “crystallinity index”. The change in the LOI reflects the crystalline and amorphous regions. HBI reflects the crystalline cellulose and intra and intermolecular regularity [25]. Crystallinity increased with the increase in the LOI, but it decreased with an increase in the HBI. The crystallization parameters based on Equations (2) and (3) and the LOI and HBI results are shown in Table 3. Native bamboo had a smaller LOI value and a larger HBI value, indicating that the raw material possessed lower crystallinity. Moreover, the LOI values for the H_2_SO_4_ and NaOH pretreated samples increased and a higher temperature gave rise to higher LOI values, indicating that the CrI values of bamboo increased after the H_2_SO_4_ and NaOH pretreatments, and the variation in the HBI values was opposite that of the LOI values. 

The d values after all of the pretreatments decreased in the order of glycerol > H_2_SO_4_ > NaOH, showing that the crystal layer spacing of cellulose after pretreatment decreased and the purity of cellulose increased. All samples exhibited typical cellulose I diffraction angles (2θ) of 16.02, 22.5, and 34.76°, which were assigned to the diffraction planes of 101, 002, and 040, respectively (Figure 3a). The variation in the diffraction angles (2θ) from 21.26 to 22.86° indicates that crystallinity changed significantly during the pretreatment [26,27]. The position of 002 exhibited an obvious offset after the pretreatment (Figure 3b). The peaks in samples after the H_2_SO_4_ and NaOH pretreatments at 135 °C moved to the right, indicating a decrease in the lattice parameters of the crystalline region and inter-planar spacing [28]. In addition, the 002 peaks of the samples in Figure 3a were high and sharp after the NaOH pretreatment, indicating that the conversion began with the formation of Na-cellulose I. Na^+^ ions destroy the inter-molecular hydrogen bonds between the lattice planes [6,7]. An interesting phenomenon that occurred in the glycerol pretreated bamboo was two irregular peaks appearing near the 040 peak (Figure 3a), but the reason for this phenomenon was unclear and requires further study.

### 3.5. CP/MAS ^13^C NMR Analysis

All samples exhibited similar NMR spectra. Therefore, the native bamboo and H_2_SO_4_ and NaOH pretreated bamboo (135 °C) were selected to compare the changes in chemical structure (Figure 4). Figure 4a shows that the signals at 173.1 and 21.3 ppm corresponded to the acetyl groups of hemicelluloses [6,7] in the native bamboo spectrum. The disappearance of the signal in the bamboo pretreated with H_2_SO_4_ and NaOH indicated that lignin and hemicellulose were removed during the pretreatment, which was consistent with the disappearance of the signal at 1730 cm^−1^ in the FT-IR spectrum.

The signals at 33.1 and 56.4 ppm were assigned to methoxyl and methylene groups of lignin; signals at 148.2, 147.9, 132.8, and 115.4 ppm were assigned to C-3 (ether-linked), C-3/5(non-ether-linked), C-1, and C-5 in guaiacyl units, respectively. The signals at 160.3 and 153.4 were assigned to C-3/5 (ether-linked) and C-1 in syringyl units, respectively. The reduction in these signals suggested that the aromatic groups of lignin in the pretreated bamboo had decreased significantly. Furthermore, the signal at 148.2 ppm in the NaOH pretreated bamboo was weaker than that of the H_2_SO_4_ pretreated bamboo, indicating that the NaOH pretreatment resulted in better delignification (Table 1).

The signal at 89.2 ppm corresponded to C-4 of crystalline cellulose and became stronger after the pretreatment, particularly the NaOH pretreatment, indicating that the cellulose content in the pretreated bamboo samples had clearly increased. The signal at 84.6 ppm was divided into two peaks of 88.7 and 83.1 ppm for the NaOH pretreated native bamboo (Figure 4b), and the signals were extensive and strong, suggesting that C4 chemical shifts occurred during the transformation from cellulose I to II [6,7]. The resonance at 105 ppm was assigned to the C-1 of cellulose and the xylans from the hemicelluloses. The signals at 63.1 and 75 ppm corresponded to the C-6 of crystalline cellulose and C-5 of cellulose and xylans, respectively, indicating higher amounts of holocellulose in the pretreated samples.

As shown in Figure 4c,d, the crystallinity of the cellulose samples was determined from the area of the crystalline cellulose C-4 signal (86–93.0 ppm) A, and the area of the mixed amorphous cellulose and xylan C4-signal (80–86 ppm), B, CrI = A/(A + B) × 100%. The peak was analyzed by Origin software (9.0), and the cumulative spectra were generated with a Gaussian shaped curve. Complex experimental data can be simulated by several separable crystalline and amorphous curves [29,30]. The CrI values of the samples were 24.83%, 34.88%, and 38.26% for the native, H_2_SO_4_, and the NaOH pretreated bamboo, respectively. The increase in CrI after the pretreatments suggested the removal of lignin, hemicelluloses, and amorphous cellulose, which was consistent with the FT-IR and XRD results. Similar findings have been reported previously [31].

### 3.6. SEM Analysis

SEM micrographs of the untreated and pretreated bamboo samples are shown in Figure 5. The uneven surface structure in the raw material was due to the existence of lignin and other extractives (Figure 5a). Hemicelluloses and lignin clustered around the cellulose in the cell walls were also visible. The diameter of the fiber bundles after the H_2_SO_4_ and NaOH pretreatments decreased from more than 50 ± 5 µm to 20 ± 2 µm (H_2_SO_4_) and less than 5 ± 2 µm (NaOH), indicating that NaOH effectively disrupted the robust structure of the raw materials relative to H_2_SO_4_. It is well known that lignin acts as a “glue” to strengthen cell walls. Cell walls will partly collapse without lignin, a highly branched and complex polymer with aromatic constituents [32]. NaOH dissolves bulk lignin and disrupts the initial fiber structure, leading to the disaggregation of micro-fibrils from their neighboring fibers [33].Therefore, the structure of the bamboo samples after the pretreatment was more ordered, delicate, and smooth, and appeared like a light emitting structure (Figure 5d–f). The highly ordered crystalline regions of fiber are comprised of several cellulose micro fibrils aligned in the fiber axis direction [6,34].

### 3.7. Thermal Stability 

The TG and DTG curves of the native materials and pretreated samples are illustrated in Figure 6. The pyrolysis characteristics are listed in Table 4. Degradation of the samples was divided into mildly degraded temperature (T1), main pyrolysis stage, maximum degraded temperature (T2), and charring temperature (T3).

All samples lost weight initially in the middle graded stage because of the evaporation of lower molecular weight compounds and loosely bound moisture at temperatures ≤ 100 °C [6,7,17,35]. Few differences in weight loss and low weight loss rates were observed in all samples (Figure 6, Table 4). The pyrolysis temperature of 279 °C was associated with the degradation of hemicelluloses, and that at 368 °C corresponded to pyrolysis of cellulose. The T2 order in all samples was NaOH > glycerol > H_2_SO_4_. The highest T2 (346 °C) in the NaOH pretreated sample showed remarkable thermal stability [6,7], possibly because the sample contained the highest percentage of cellulose (Table 1). However, the relatively high T2 of the glycerol-pretreated sample might be due to high ash content (0.15%). Most hemicelluloses and cellulose decomposed when the temperature reached 400 °C, and the remaining were ash residue and lignin. Therefore, the high ash residue could be attributed to incomplete pyrolysis of lignin.

### 3.8. Enzymatic Hydrolysis of the NaOH Pretreated Bamboo Samples

Figure 7a,b indicates that the glucose and xylose yields of the NaOH pretreated bamboo increased significantly with cellulase loading from 5 to 10 FPU/g DM. The hydrolysis yield of cellulose in NaOH pretreated bamboo at 135 °C increased from 18.7% to 98.1% and the xylose yield increased from 19.6% to 94.4%, respectively. NaOH pretreated bamboo with relatively low lignin content was hydrolyzed with CEL and/or XYL to investigate the effect of removing lignin and solubilizing xylan on the bamboo fractions during hydrolysis. Some cellulose was hydrolyzed by XYL alone and the hydrolysis yields of cellulose increased with the increase in XYL concentration.

The hydrolysis yield of cellulose in native bamboo and pretreated bamboo (117 °C and 135 °C) by 1 mg protein/g DM XYL was 2.51–43.3%, indicating that some cellulolytic activities were present in the commercial XYL preparation, particularly in the 135 °C pretreated bamboo. After supplementing with 0.5 CEL to 1 mg protein/g DM XYL, the hydrolysis yield of NaOH pretreated bamboo at 135 °C increased from 84.2% to 98.1%. The hydrolysis yield of cellulose in NaOH pretreated bamboo further increased to 100.1%when using 1mg protein/g DM XYL with CEL. These results indicate that XYL supplementation was more effective than increasing cellulase loading during hydrolysis of the NaOH pretreated bamboo. The XYL preparation hydrolyzed residual xylan in the NaOH pretreated bamboo and increased the accessibility of cellulase to cellulose, which was consistent with a previous report regarding the synergy between CEL and XYL on the hydrolysis of steam pretreated corn stover and hydrothermally pretreated wheat straw [36].

The reducing sugars released by the NaOH pretreated bamboo samples increased to different degrees with increasing dosages of CEL. These results were in good agreement with previous studies [8,37].

## 4. Conclusions

We carried out a comparative study to pretreat bamboo with NaOH, H_2_SO_4_, and glycerol. The structural and chemical analysis results revealed that the NaOH pretreatment using a sand bath was the most effective method to compare H_2_SO_4_ with glycerol. The NaOH pretreatment resulted in uniform delignification, defibrillation, maximum CrI, and preferable thermo-stability. Decreases in diameter of the cellulose fibers from 50 ± 5 to 5 ± 2 µm and a DP of cellulose from 396 to 329 were observed. Supplementing with xylanase was more effective than increasing cellulase loading for scarification of the pretreated bamboo fractions, particularly after the NaOH pretreatment, demonstrating the positive effect of the NaOH pretreated bamboo in a sand bath with added xylanase for improving hydrolysis of bamboo. 

## Figures and Tables

**Figure 1 materials-12-02570-f001:**
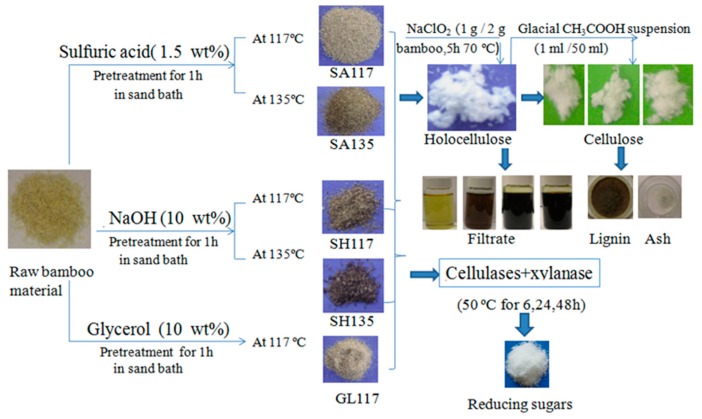
Scheme for preparing the bamboo samples and isolating cellulose, hemicelluloses, and lignin.

**Figure 2 materials-12-02570-f002:**
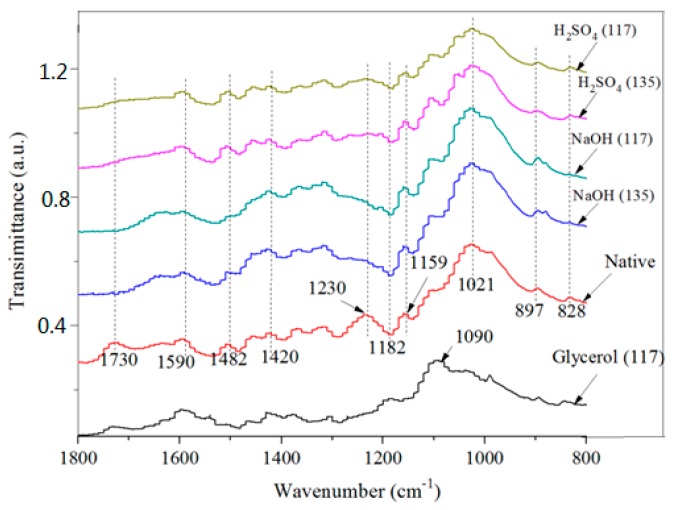
Infrared spectra of bamboo substrates before and after the pretreatments.

**Figure 3 materials-12-02570-f003:**
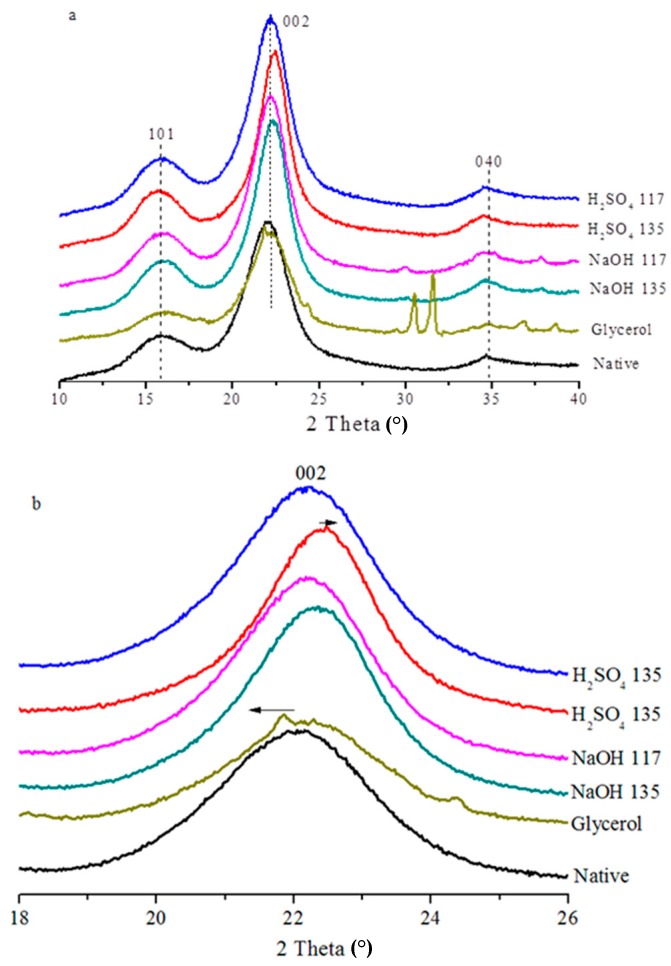
X-ray diffraction patterns of untreated and pretreated bamboo solids (**a**), and layer spacing variation for the 002 crystal plane diffraction peak (**b**).

**Figure 4 materials-12-02570-f004:**
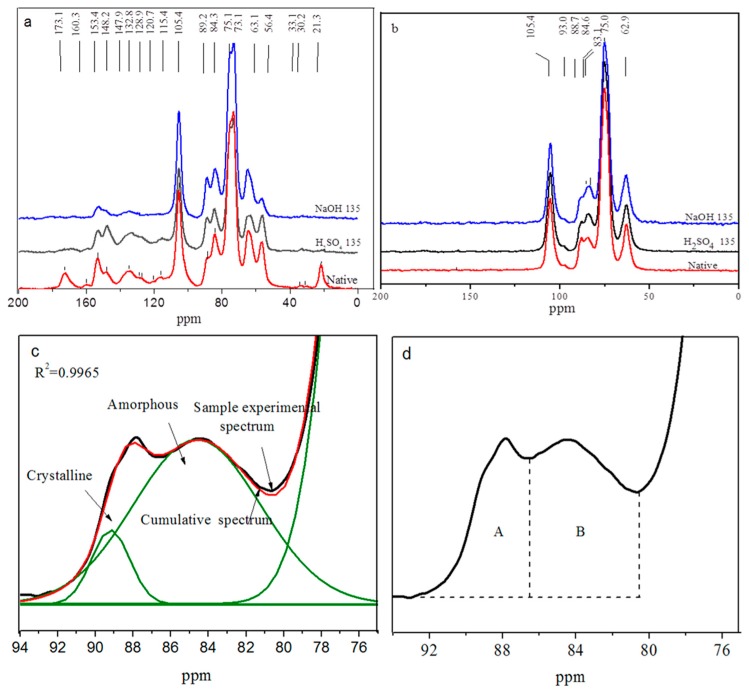
Solid-state cross polarization/magic angle spinning (CP/MAS)^13^C nuclear magnetic resonance (NMR) spectra of solid particle samples (native, SA135, and SH135) (**a**), holocellulose samples for raw bamboo materials and pretreated (naive, SA135, and SH135) (**b**); the fitted curve of the C4 region (**c**), and the crystalline calculation of the C4 region (**d**).

**Figure 5 materials-12-02570-f005:**
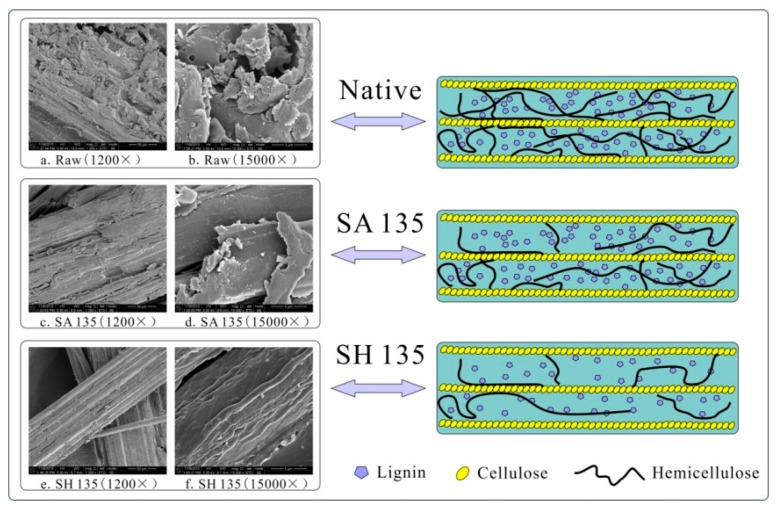
Scanning electron microscopic images of raw bamboo and pretreated bamboo samples. Note: (**a**) Native (raw bamboo materials) (1200×, 50 µm); (**b**) native (15,000×, 5 µm); (**c**) SA135 (pretreated bamboo with H_2_SO_4_ at 135 °C) (1200×, 50 µm); (**d**) SA135 (15,000×, 5 µm); (**e**) SH135 (pretreated bamboo with NaOH at 135 °C) (1200×, 50 µm); (**f**) SH135 (15,000×, 5 µm).

**Figure 6 materials-12-02570-f006:**
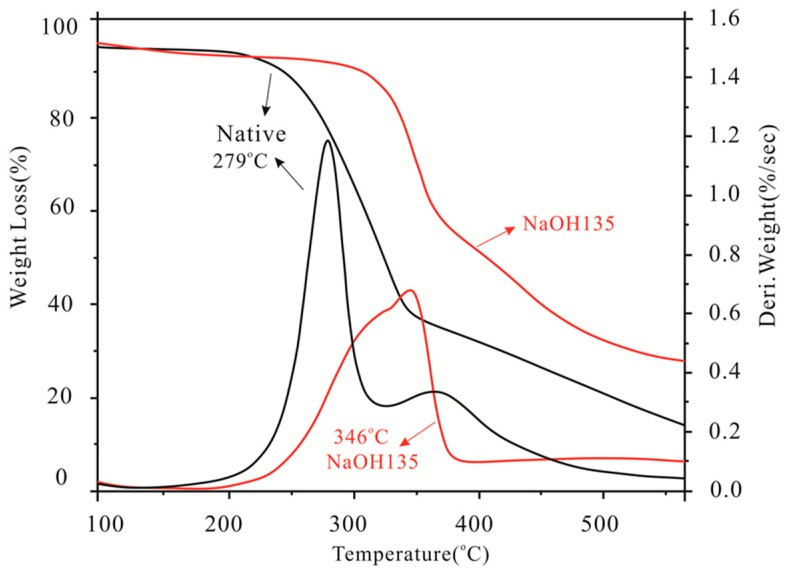
Thermogravimetry of the pretreated bamboo and native bamboo samples.

**Figure 7 materials-12-02570-f007:**
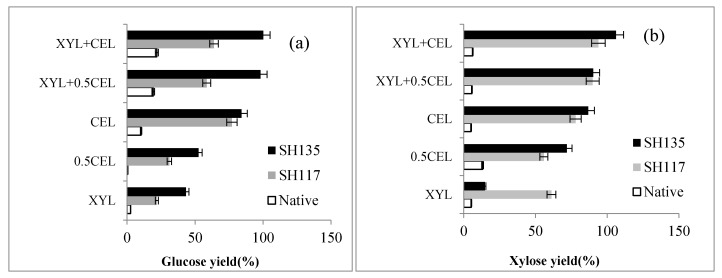
Hydrolysis of 2% pretreated bamboo by CEL (5, 10 FPU of Celluclast 1.5 L/g DM and 500 nkat of Novozyme 188/g DM) and/or XYL (2 mg protein/g DM) at 50 °C and pH 5.0 for 48 h (**a,****b**). Error bars represent standard errors of two independent experiments.

**Table 1 materials-12-02570-t001:** Chemical composition of the bamboo samples with/without pretreatment.

Pretreatment	Temperature °C	Holocellulose (%)	Cellulose (%)	Hemicellulose (%)	Lignin (%)	Ash (%)	Extractives (%)
Native	-	60.45 ± 5.0	46.94 ± 0.1	15.59 ± 0.4	25.96 ± 0.3	8.39 ± 3.9	11.54 ± 0.9
Glycerol	117	67.53 ± 0.8	51.94 ± 0.2	15.59 ± 0.6	18.39 ± 0.1	9.54 ± 3.5	3.19 ± 0.1
H_2_SO_4_	117	77.01 ± 0.6	57.12 ± 0.3	19.89 ± 0.3	15.68 ± 1.7	5.99 ± 0.0	1.87 ± 0.2
	135	78.07 ± 0.3	61.24 ± 0.2	16.83 ± 0.1	14.71 ± 2.9	5.11 ± 0.0	2.11 ± 0.1
NaOH	117	82.13 ± 0.7	68.15 ± 0.4	13.98 ± 0.3	13.29 ± 0.2	1.48 ± 2.5	2.66 ± 0.3
	135	87.38 ± 0.3	73.94 ± 0.1	14.44 ± 0.2	8.93 ± 0.6	3.12 ± 1.9	1.03 ± 0.1

**Table 2 materials-12-02570-t002:** Viscosity and degree of polymerization (DP) values of the bamboo samples with/without pretreatment.

Pretreatment	Temperature	DP	η (mL/g)
Native	-	396	132.62
Glycerol	117 °C	362	121.63
H_2_SO_4_	117 °C	372	129.62
	135 °C	363	122.96
NaOH	117 °C	336	110.67
	135 °C	329	103.88

**Table 3 materials-12-02570-t003:** Crystal values of raw and pretreated bamboo samples.

Pretreatment	Temperature	2θ (°)	C_r_I (%)	d (nm)	LOI ^a^	HBI ^b^
Native	-	21.26	36.91	0.40	0.70	1.05
Glycerol	117 °C	22.28	38.09	0.39	0.74	1.04
H_2_SO_4_	117 °C	22.48	42.66	0.39	0.84	1.01
	135 °C	22.20	40.63	0.38	0.71	1.01
NaOH	117 °C	22.86	48.42	0.36	0.91	1.00
	135 °C	22.19	46.50	0.38	0.88	0.99

^a^ Lateral order index; ^b^ hydrogen bond intensity.

**Table 4 materials-12-02570-t004:** Pyrolysis characteristics of the native material and pretreatment bamboo samples.

Pretreatment	Temperature	The Mild Degraded Stage		The Main Pyrolysis Stage		The Charring Stage	
		T_1_ ^a^	WL ^d^ (%/°C)	T_2_ ^b^	WL (%/S)	T_3_ ^c^	Ar ^e^ (%/°C)
Native	-	210	0.29	279 (368) ^f^	1.18	365	0.33
Glycerol	117 °C	212	0.25	310	0.88	369	0.15
H_2_SO_4_	117 °C	194	0.21	312	0.62	356	0.13
	135 °C	215	0.2	325	0.79	405	0.13
NaOH	117 °C	235	0.14	346	0.69	439	0.12
	135 °C	230	0.26	330	1.28	383	0.15

^a^ Mildly degraded temperature; ^b^ Main pyrolysis stage, maximum degraded temperature; ^c^ Charring temperature; ^d^ Weight loss rate; ^e^ Ashing reside rate; ^f^ Lower separated peaks.

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
