# Peer review of "Influence of Different Pretreatments on the Structure and Hydrolysis Behavior of Bamboo: A Comparative Study"

_materials, 2019, doi:10.3390/ma12162570_

Round 1
Reviewer 1 Report
Interesting and excellent work. It may be needs a little more connection with biodiesel or biofuels or with what is the final target and the key point of this study. Some typing errors (e.g. space characters line 24, 75, 81, 82) need to be corrected.
Author Response
Replies to Reviewer 1
Interesting and excellent work. It may be needs a little more connection with biodiesel or biofuels or with what is the final target and the key point of this study. Some typing errors (e.g. space characters line 24, 75, 81, 82) need to be corrected.
Response: Space characters line in the manuscript has been revised and final target and the key point of this study has been added in line 88-90.ou
Thank you.
Reviewer 2 Report
This manuscript addresses the influence of different pretreatment methods (NaOH, H2SO4 and glycerol solutions) on the structural properties of bamboo fibers. Many analytical methods were applied in this study, including XRD, SEM, FT-IR. Based on the results the authors concluded that the most effective pretreatment was the one with the use of NaOH, and for such pretreated biomass, the series of enzymatic hydrolysis tests were performed.
The subject of this paper is quite interesting but in the manuscript is full of mistakes and inconsistencies, and in general, carelessly prepared which makes it difficult to read in some parts. I think that in present form this paper is not acceptable, and Authors needs to improve it if significantly and rewrite in some parts.
General comments:
There is a number of spacing errors, typos, and formatting errors in the manuscript. The whole manuscript needs to be carefully read and all of the mistakes have to be corrected.
A lot of references that are included in the main text are not listed in the “References” section.
The number of replications of each experiment should be added, and the statistical analysis of obtained results is highly recommended.
In the introduction, it would be beneficial to include the short part about what kind of information can be obtained after each type of analyses used in this study.
The novelty of this work should be better explained.
Specific comments:
I will start with the title – there is a mistake in it. „Influence of different pretreatments on the structural and hydrolysis behavior of bamboo: A comparative study”. I think it should be “structure” or “structural properties”.
Introduction
L51 – the authors wrote “The main aim of this study was to study the influence of acid, alkaline (NaOH and H2SO4),…” it should be “…acid (H2SO4), alkaline (NaOH),…”
Materials and Methods
L68 and 73 – “A bamboo sample (100 g oven‐dry basis) was used for each pretreatment experiment.” and “about 10 g of bamboo powder (dry weight) was pretreated”. So, 10g or 100g of the bamboo sample was used in experiments?
L69 – could you add the information about what kind (name, manufacturer) of xylanase preparation was applied in this study?
L73 – could you justify the choice of reagents concentration used for pretreatment? In my opinion, it is difficult to compare 1.5% acid with 10% alkali – the difference in concentration is too high.
L80-101 – “2.3. Components analysis” – the whole subsection should be checked as there is a lot of inconsistencies. The references should be checked and updated, for example in the cited paper (Xie et al. 2014) only methods for lignin determination are included, while the text shows that entire methodology was based on this publication.
Also, the authors wrote “Lignin content was fractionated into acid-insoluble and acid-soluble lignin”, but no results regarding acid-soluble lignin are included.
L141-155 – if the enzymatic hydrolysis was performed in a total volume of 3 mL with 2% solid loading it gives 0.06g of the pretreated bamboo added to the sample – do authors think that this volume is sufficient to perform reliable experiments?
Also, how were the individual sugars (glucose, xylose) content analyzed? I could not find any reference in the text, but in the “references” section, the Millers’ method is listed. However, using this method the sum of reducing substances can be obtained. And this leads to the second question how was the percentage of cellulose to glucose and xylan to xylose conversion calculated? Also, in equation 5, there is a mistake – the coefficient should be 0.88, not 0.9 (0.9 is for cellulose to glucose)
L169 – “The quantity of lignin decreased after pretreatment, and the order of lignin content was glycerol<H2SO4
L169-172 – it would be beneficial to include a table with obtained results.
Also, when authors wrote “The DP value after pretreatment decreased slightly compared with that of native bamboo from 396 to 329 (Li et al.2010 b).” did you mean your own results of the one obtained by Li et al? (Speaking of, “Li et al 2010b” is also not included in the references)
L202-213 – Could you add some references to explain why the pretreatments result in higher CrI?
L2017 – Table 2 – the meaning of “a” and “b” (in superscript) should be explained in the table footer
L236-238 – “The peaks in samples after the H2SO4 and NaOH pretreatments at 135°C moved to the right, indicating a decrease in the lattice parameters of the crystalline region and inter‐planar spacing (Hall et al. 2010).” Could you please explain why the peak in the sample after the glycerol pretreatment at 135°C moved to the left? (Hall et al 2010 is also not listed)
L300 – Figure 5 – the legend does not match the symbols in the figure
L310 – Figure 6 – this figure is hard to follow – according to the description in the text it should be divided into a and b
L355 – Figure 7 – why is such a big difference in % of glucose yield in Fig 7b and 7d? If I understand it correctly, the glucose yield at 10 FPU (fig 7d) should be comparable with the yield after 48h in fig 7b, while there is a big difference between these results (86vs 62?%).

Author Response
Replies to Reviewer 2
Specific comments:
I will start with the title – there is a mistake in it. „Influence of different pretreatments on the structural and hydrolysis behavior of bamboo: A comparative study”. I think it should be “structure” or “structural properties”.
Response: it has been revised.
Introduction
L51 – the authors wrote “The main aim of this study was to study the influence of acid, alkaline (NaOH and H2SO4),…” it should be “…acid (H2SO4), alkaline (NaOH),…”
Response: it has been revised.
Materials and Methods
L68 and 73 – “A bamboo sample (100 g oven‐dry basis) was used for each pretreatment experiment.” and “about 10 g of bamboo powder (dry weight) was pretreated”. So, 10g or 100g of the bamboo sample was used in experiments?
Response: The experiment used a total of 100g bamboo powder, of which 10g was used for each pretreatment.
L69 – could you add the information about what kind (name, manufacturer) of xylanase preparation was applied in this study?
Response: it has been revised in line 103.
L73 – could you justify the choice of reagents concentration used for pretreatment? In my opinion, it is difficult to compare 1.5% acid with 10% alkali – the difference in concentration is too high.
Response: The concentration of dilute acids and alkali has been determined in the previous article,eg:referece [2]
L80-101 – “2.3. Components analysis” – the whole subsection should be checked as there is a lot of inconsistencies. The references should be checked and updated, for example in the cited paper (Xie et al. 2014) only methods for lignin determination are included, while the text shows that entire methodology was based on this publication.
Also, the authors wrote “Lignin content was fractionated into acid-insoluble and acid-soluble lignin”, but no results regarding acid-soluble lignin are included.
Response: Components analysis was based on US Renewable Energy Laboratory (NREL, USA)
L141-155 – if the enzymatic hydrolysis was performed in a total volume of 3 mL with 2% solid loading it gives 0.06g of the pretreated bamboo added to the sample – do authors think that this volume is sufficient to perform reliable experiments?
Response: Hydrolysis method and reagent dosage according to the method provided by the US Renewable Energy Laboratory (NREL, USA)
Also, how were the individual sugars (glucose, xylose) content analyzed? I could not find any reference in the text, but in the “references” section, the Millers’ method is listed. However, using this method the sum of reducing substances can be obtained. And this leads to the second question how was the percentage of cellulose to glucose and xylan to xylose conversion calculated? Also, in equation 5, there is a mistake – the coefficient should be 0.88, not 0.9 (0.9 is for cellulose to glucose)
Response: it has been revised.
L169 – “The quantity of lignin decreased after pretreatment, and the order of lignin content was glycerol<H2SO4
Response: it has been revised.
L169-172 – it would be beneficial to include a table with obtained results.
Also, when authors wrote “The DP value after pretreatment decreased slightly compared with that of native bamboo from 396 to 329 (Li et al.2010 b).” did you mean your own results of the one obtained by Li et al? (Speaking of, “Li et al 2010b” is also not included in the references)
Response: it has been revised. Li et al.2010 has added in the paper.
L202-213 – Could you add some references to explain why the pretreatments result in higher CrI?
Response: it has been explained in some references.
L2017 – Table 2 – the meaning of “a” and “b” (in superscript) should be explained in the table footer.
Response: it has been revised.
L236-238 – “The peaks in samples after the H2SO4 and NaOH pretreatments at 135°C moved to the right, indicating a decrease in the lattice parameters of the crystalline region and inter‐planar spacing (Hall et al. 2010).” Could you please explain why the peak in the sample after the glycerol pretreatment at 135°C moved to the left? (Hall et al 2010 is also not listed)
Response: it has been revised in 391-397.
L300 – Figure 5 – the legend does not match the symbols in the figure
Response: it has been revised.
L310 – Figure 6 – this figure is hard to follow – according to the description in the text it should be divided into a and b
Response: The figure contains two indicators, each of which contains two lines of pre-processed and without pre-processing. Different indicators correspond to different colors.
L355 – Figure 7 – why is such a big difference in % of glucose yield in Fig 7b and 7d? If I understand it correctly, the glucose yield at 10 FPU (fig 7d) should be comparable with the yield after 48h in fig 7b, while there is a big difference between these results (86vs 62?%).
Response: The amount of enzyme in Figure 7b is 10FPU, and the amount of enzyme in Figure 7d is 20FPU. Therefore, the difference in glucose yield after hydrolysis is large.
We would like to express our sincere thanks to the reviewers for the constructive and positive comments.
I look forward to hearing from you soon.
With best wishes,
Yours sincerely,
Jie Chu
7-24-2019
Round 2
Reviewer 2 Report
The manuscript was improved compared to the original version, and some of my previous suggestions (but not all of them) have been included into the text or explained. However, I still think that this manuscript needs to be improved prior to publication in Materials journal.
There is still a lot of spacing errors in the manuscript – a lot of then has been corrected but also a lot of them remained. Also, still not all references are listed in the “References” section (e.g. L49 – Sun [12], L91 – Peng and She, L355 – Ma et al 2013), or wrong paper is cited (L89 – it should be NREL protocol)
The authors didn’t response to my comment concerning a number of replications made and statistical analysis. I think that it should be included in the “Materials and methods” section.
Unfortunately, also the authors didn’t respond or respond poorly on some of my previous comments.
1. From the previous round of review: L68 and 73 – “A bamboo sample (100 g oven‐dry basis) was used for each pretreatment experiment.” and “about 10 g of bamboo powder (dry weight) was pretreated”. So, 10g or 100g of the bamboo sample was used in experiments? Response: The experiment used a total of 100g bamboo powder, of which 10g was used for each pretreatment.
I think that more accurate would be “A bamboo total sample of 100 g oven‐dry basis was used for pretreatment experiment”, or something else but clearly defining that 100 g was in total and 10 g was used for each pretreatment. In the present form it is confusing.
2. From the previous round of review: L73 – could you justify the choice of reagents concentration used for pretreatment? In my opinion, it is difficult to compare 1.5% acid with 10% alkali – the difference in concentration is too high. Response: The concentration of dilute acids and alkali has been determined in the previous article,eg:referece [2]
I’m confused, in the mentioned paper ([2] - Li et al 2016), the comparison between SAA, 1% H2SO4 and 1% NaOH was presented, and I don’t understand how it could justify a choice of 1.5% H2SO4 and 10% NaOH applied in this paper.
3. From the previous round of review: L80-101 – “2.3. Components analysis” – the whole subsection should be checked as there is a lot of inconsistencies. The references should be checked and updated, for example in the cited paper (Xie et al. 2014) only methods for lignin determination are included, while the text shows that entire methodology was based on this publication. Also, the authors wrote “Lignin content was fractionated into acid-insoluble and acid-soluble lignin”, but no results regarding acid-soluble lignin are included.
Response: Components analysis was based on US Renewable Energy Laboratory (NREL, USA)
The subsection “Component analysis” is still not clear for me. If, as the authors wrote in the response to my previous comments, the whole compositional analysis was performed according to NREL methodology, then I don’t understand why the description in lines 89-99 is included. According to NREL methodology the structural carbohydrates and lignin (TP-510-42618), extractives (TP-510-42619) and ash (TP-510-42622) can be determined.
If the procedure included in L89-99 is to describe the holocellulose and cellulose content determination, the a proper reference should be included (in cited paper – Peng and She, 2014 – I couldn’t find such procedure, especially that this is a review paper), and the information that holocellulose and cellulose was determined according to NREL (L88) should be deleted.
Also, the cited reference regarding NREL protocols is improper (review paper not the protocol)
4. From the previous round of review: Also, how were the individual sugars (glucose, xylose) content analyzed? I could not find any reference in the text, but in the “references” section, the Millers’ method is listed. However, using this method the sum of reducing substances can be obtained. And this leads to the second question how was the percentage of cellulose to glucose and xylan to xylose conversion calculated? Also, in equation 5, there is a mistake – the coefficient should be 0.88, not 0.9 (0.9 is for cellulose to glucose). Response: it has been revised.
The equation was corrected, but other issues were not addressed. In my opinion, it is necessary to specify the method used for sugar determination.
5. From the previous round of review: Also, when authors wrote “The DP value after pretreatment decreased slightly compared with that of native bamboo from 396 to 329 (Li et al.2010 b).” did you mean your own results of the one obtained by Li et al?
In the revised paper, there is the same issue “The DP value after pretreatment decreased slightly compared with that of native bamboo from 396 to 329 [23]” (L190-191). My question is, why the reference is included if authors wrote about their own results?
6. From the previous round of review: L202-213 – Could you add some references to explain why the pretreatments result in higher CrI? Response: it has been explained in some references.
Could you please include some of them (the references you meant) in the manuscript along with a short description. The information that someone explained it somewhere is, in my opinion, insufficient.
7. From the previous round of review: L355 – Figure 7 – why is such a big difference in % of glucose yield in Fig 7b and 7d? If I understand it correctly, the glucose yield at 10 FPU (fig 7d) should be comparable with the yield after 48h in fig 7b, while there is a big difference between these results (86vs 62?%). Response: The amount of enzyme in Figure 7b is 10FPU, and the amount of enzyme in Figure 7d is 20FPU. Therefore, the difference in glucose yield after hydrolysis is large.
I still have some doubts regarding this part. Please correct me if I’m wrong, but in the first set of experiments Authors were analyzing the effect of hydrolysis time (6, 24, 48h) with constant enzyme dose (10 FPU) on the efficiency of the process (L367-368) and the maximum obtained glucose yield was 86.5% for sample SH135 (L352, Fig 7b 3rd black bar). In the second set of experiment, authors were conduction hydrolysis for constant time (48h) with different enzyme dose (5, 10,20, 40 FPU/ g DM) (L 368-369). The results presented in Fig 7d indicates, that for sample SH135 (black bars), the glucose yield was approx. 20% (5 FPU), approx. 62^ (10 FPU), approx. 70% (20 FPU) and approx. 98% (40 FPU). My question is why, when comparing the 3rd black bar in FIG 7b (10 FPU, 48h, SH135; glucose yield 86%) with second black bar in Fig 7d (48h, 10 FPU,SH135; glucose yield 62%) the difference between these results is so high – in my opinion, they should be quite similar as the same conditions were applied (10 FPU, 48h, sample SH135).
Referring to the authors response to this question after 1st round of review, that “The amount of enzyme in Figure 7b is 10FPU, and the amount of enzyme in Figure 7d is 20FPU. Therefore, the difference in glucose yield after hydrolysis is large”, it is also not clear as the yield for 20 FPU (Fig 7d) is lower than for 10 FPU (Fig 7b).
What is more, I found few more things to correct:
- L175 – the reference was lost – only square bracket remained
- L181-182 – authors wrote “The high amounts of ash after the glycerol pretreatment could represent some ash and non-structural carbohydrates, starch, wax, fats, tannins, and pigments in the glycerol”. I think this sentence should be rewritten as it indicates that the glycerol solution used in the pretreatment is a source of all of these compounds.
- L207 – word “intermolecular” is repeated
- L218 – “Table 2” should be Table 3
- L345 – why is a reference included here if this sentence is connected with own results?
- L374-376 - “Figure 7(c,d) indicates that the glucose and xylose yields of the NaOH pretreated bamboo increased slightly with cellulase loading from 5 to 40 FPU/g DM. The hydrolysis yield of cellulose in NaOH pretreated bamboo at 135°Cincreased from 18.7% to 98.1%”. An increase from 19 to 98% is significant, not slight.
- L378-380 – the description does not match to the Figure
- L 382-398 – Authors use two types of sample description – in methodology and in figure captions “CEL (5, 10 FPU of Celluclast 1.5 L/g DM and 500 nkat of Novozyme 188/g DM) and/or XYL (2 mg protein/g DM)”, while in the main text authors use “0.5 CEL to 1 mg protein/g DM XYL” and “mg protein/g DM XYL with CEL” (L157-158, L370-371 vs. L388, 390, 393). It is highly confusing, I think it should be uniformed.
Finally, I suggest authors ask a native English speaking colleague for help with their manuscript.
Author Response
Response to referee’s comments
The manuscript was improved compared to the original version, and some of my previous suggestions (but not all of them) have been included into the text or explained. However, I still think that this manuscript needs to be improved prior to publication in Materials journal.
RE: We would like to thank the referee for his comments and suggestions. We have revised the manuscript as per the suggestions and comments.
There is still a lot of spacing errors in the manuscript – a lot of then has been corrected but also a lot of them remained. Also, still not all references are listed in the “References” section (e.g. L49 – Sun [12], L91 – Peng and She, L355 – Ma et al 2013), or wrong paper is cited (L89 – it should be NREL protocol)
RE: Thanks for the suggestion and pointing out the mistake. We have corrected the mistake.
The authors didn’t response to my comment concerning a number of replications made and statistical analysis. I think that it should be included in the “Materials and methods” section.
Unfortunately, also the authors didn’t respond or respond poorly on some of my previous comments.
From the previous round of review: L68 and 73 – “A bamboo sample (100 g oven‐dry basis) was used for each pretreatment experiment.” and “about 10 g of bamboo powder (dry weight) was pretreated”. So, 10g or 100g of the bamboo sample was used in experiments? Response: The experiment used a total of 100g bamboo powder, of which 10g was used for each pretreatment.
I think that more accurate would be “A bamboo total sample of 100 g oven‐dry basis was used for pretreatment experiment”, or something else but clearly defining that 100 g was in total and 10 g was used for each pretreatment. In the present form it is confusing.
RE: First we would like convey our sincere apologies for not considered your earlier suggestions. Now we have revised these sentences as per your suggestions.
From the previous round of review: L73 – could you justify the choice of reagents concentration used for pretreatment? In my opinion, it is difficult to compare 1.5% acid with 10% alkali – the difference in concentration is too high.
RE: Thanks for suggestions: We have followed the methodology used by earlier paper (reference 19).
I’m confused, in the mentioned paper ([2] - Li et al 2016), the comparison between SAA, 1% H2SO4and 1% NaOH was presented, and I don’t understand how it could justify a choice of 1.5% H2SO4and 10% NaOH applied in this paper.
From the previous round of review: L80-101 – “2.3. Components analysis” – the whole subsection should be checked as there is a lot of inconsistencies. The references should be checked and updated, for example in the cited paper (Xie et al. 2014) only methods for lignin determination are included, while the text shows that entire methodology was based on this publication. Also, the authors wrote “Lignin content was fractionated into acid-insoluble and acid-soluble lignin”, but no results regarding acid-soluble lignin are included.
RE: We would like to thank the referee for the comments:
We have followed the complete methodology mentioned by US Renewable Energy Laboratory (NREL, USA)
The subsection “Component analysis” is still not clear for me. If, as the authors wrote in the response to my previous comments, the whole compositional analysis was performed according to NREL methodology, then I don’t understand why the description in lines 89-99 is included. According to NREL methodology the structural carbohydrates and lignin (TP-510-42618), extractives (TP-510-42619) and ash (TP-510-42622) can be determined.
If the procedure included in L89-99 is to describe the holocellulose and cellulose content determination, the a proper reference should be included (in cited paper – Peng and She, 2014 – I couldn’t find such procedure, especially that this is a review paper), and the information that holocellulose and cellulose was determined according to NREL (L88) should be deleted.
Also, the cited reference regarding NREL protocols is improper (review paper not the protocol)
RE: We would like to thank the referee for the comments:
Thanks for suggestions: We have followed the methodology used by earlier paper (reference 19).And the information that holocellulose and cellulose was determined according to NREL (L88) should had been deleted.
From the previous round of review: Also, how were the individual sugars (glucose, xylose) content analyzed? I could not find any reference in the text, but in the “references” section, the Millers’ method is listed. However, using this method the sum of reducing substances can be obtained. And this leads to the second question how was the percentage of cellulose to glucose and xylan to xylose conversion calculated? Also, in equation 5, there is a mistake – the coefficient should be 0.88, not 0.9 (0.9 is for cellulose to glucose). Response: it has been revised.
The equation was corrected, but other issues were not addressed. In my opinion, it is necessary to specify the method used for sugar determination.
RE: We would like to thank the referee for the comments:
Thanks for suggestions: We have described sugar determination followed your advice in L157-169,thank you.
From the previous round of review: Also, when authors wrote “The DP value after pretreatment decreased slightly compared with that of native bamboo from 396 to 329 (Li et al.2010 b).” did you mean your own results of the one obtained by Li et al?
In the revised paper, there is the same issue “The DP value after pretreatment decreased slightly compared with that of native bamboo from 396 to 329 [23]” (L190-191). My question is, why the reference is included if authors wrote about their own results?
RE: We would like to thank the referee for the comments:
The reference is included is that consists with our results.
From the previous round of review: L202-213 – Could you add some references to explain why the pretreatments result in higher CrI? Response: it has been explained in some references.
Could you please include some of them (the references you meant) in the manuscript along with a short description. The information that someone explained it somewhere is, in my opinion, insufficient.
RE: We would like to thank the referee for the comments:
We have added reference to explain why the pretreatments result in higher CrI and explained with with a short description in Line 232-233 of paper .
From the previous round of review: L355 – Figure 7 – why is such a big difference in % of glucose yield in Fig 7b and 7d? If I understand it correctly, the glucose yield at 10 FPU (fig 7d) should be comparable with the yield after 48h in fig 7b, while there is a big difference between these results (86vs 62?%). Response: The amount of enzyme in Figure 7b is 10FPU, and the amount of enzyme in Figure 7d is 20FPU. Therefore, the difference in glucose yield after hydrolysis is large.
I still have some doubts regarding this part. Please correct me if I’m wrong, but in the first set of experiments Authors were analyzing the effect of hydrolysis time (6, 24, 48h) with constant enzyme dose (10 FPU) on the efficiency of the process (L367-368) and the maximum obtained glucose yield was 86.5% for sample SH135 (L352, Fig 7b 3rd black bar). In the second set of experiment, authors were conduction hydrolysis for constant time (48h) with different enzyme dose (5, 10,20, 40 FPU/ g DM) (L 368-369). The results presented in Fig 7d indicates, that for sample SH135 (black bars), the glucose yield was approx. 20% (5 FPU), approx. 62^ (10 FPU), approx. 70% (20 FPU) and approx. 98% (40 FPU). My question is why, when comparing the 3rd black bar in FIG 7b (10 FPU, 48h, SH135; glucose yield 86%) with second black bar in Fig 7d (48h, 10 FPU,SH135; glucose yield 62%) the difference between these results is so high – in my opinion, they should be quite similar as the same conditions were applied (10 FPU, 48h, sample SH135).
Referring to the authors response to this question after 1st round of review, that “The amount of enzyme in Figure 7b is 10FPU, and the amount of enzyme in Figure 7d is 20FPU. Therefore, the difference in glucose yield after hydrolysis is large”, it is also not clear as the yield for 20 FPU (Fig 7d) is lower than for 10 FPU (Fig 7b).
What is more, I found few more things to correct:
- L175 – the reference was lost – only square bracket remained
- L181-182 – authors wrote “The high amounts of ash after the glycerol pretreatment could represent some ash and non-structural carbohydrates, starch, wax, fats, tannins, and pigments in the glycerol”. I think this sentence should be rewritten as it indicates that the glycerol solution used in the pretreatment is a source of all of these compounds.
- L207 – word “intermolecular” is repeated
- L218 – “Table 2” should be Table 3
- L345 – why is a reference included here if this sentence is connected with own results?
- L374-376 - “Figure 7(c,d) indicates that the glucose and xylose yields of the NaOH pretreated bamboo increased slightly with cellulase loading from 5 to 40 FPU/g DM. The hydrolysis yield of cellulose in NaOH pretreated bamboo at 135°Cincreased from 18.7% to 98.1%”. An increase from 19 to 98% is significant, not slight.
- L378-380 – the description does not match to the Figure
- L 382-398 – Authors use two types of sample description – in methodology and in figure captions “CEL (5, 10 FPU of Celluclast 1.5 L/g DM and 500 nkat of Novozyme 188/g DM) and/or XYL (2 mg protein/g DM)”, while in the main text authors use “0.5 CEL to 1 mg protein/g DM XYL” and “mg protein/g DM XYL with CEL” (L157-158, L370-371 vs. L388, 390, 393). It is highly confusing, I think it should be uniformed.
RE: We would like to thank the referee for the Sincere advise :
According to your opinions, we have carried out two experiments to verify the data, and discussed with the research group. It is found that there are large errors in Figures. It is very difficult to adjust the structure within 5 days according to our experience and the content of the paper. Finally, two more accurate tables are retained in paper(part 3.8)
Finally, I suggest authors ask a native English speaking colleague for help with their manuscript.
RE: We would like to thank the referee for the Sincere advise :
A native English speaking colleague who is Pro.Aunj has edited the manuscript.

Round 3
Reviewer 2 Report
The manuscript was much improved but I still have some suggestions:
The authors still didn’t response to my comment concerning a number of replications made and statistical analysis. I think that it should be included in the “Materials and methods” section. Please update the subsection 2.10. (enzymatic hydrolysis), since authors removed the part concerning the influence of enzyme dose and hydrolysis time on the enzymatic hydrolysis from the “Results and discussion” section, the same must be done with “Materials and methods” section (e.g. L 151-152) In abstract (L15) the diameter of the cellulose fibers was accidentally deleted L187 – should be “Klason” L231 – there is “CI”, should be “CrI” L283-284 – there is “there duction”, should be (?) “the reduction”
Author Response
Dear professor,
We would like to thank the referee for your comments and suggestions.
The authors still didn’t response to my comment concerning a number of replications made and statistical analysis.
I think that it should be included in the “Materials and methods” section.
response: We would like to thank the referee for his comments and suggestions. A number of replications made and statistical analysis we have revised the manuscript .According to our understanding, it is generally the writing of these methods and materials in hydrolysis articles.
Thanks for the suggestion very much.
Please update the subsection 2.10. (enzymatic hydrolysis), since authors removed the part concerning the influence of enzyme dose and hydrolysis time on the enzymatic hydrolysis from the “Results and discussion” section, the same must be done with “Materials and methods” section (e.g. L 151-152)
Thanks for the suggestion and pointing out the mistake. We have removed hydrolysis time from the “Materials and methods” section
In abstract (L15) the diameter of the cellulose fibers was accidentally deleted L187 – should be “Klason”
First we would like convey our sincere apologies for not considered your earlier suggestions. Now we have revised these sentences as per your suggestions.
L231 – there is “CI”, should be “CrI”.
We would like to thank the referee for the comments: We have corrected the mistake.
L283-284 – there is “there duction”, should be (?) “the reduction”.
Thanks for the suggestion and pointing out the mistake. We have corrected the mistake.
Thanks for the suggestion very much.
